# The Effect of *Aspergillus flavus* on Seedling Development in Maize

**DOI:** 10.3390/plants14071109

**Published:** 2025-04-02

**Authors:** Isabella Mazzoleni, Elena Novarina, Yuki Michelangelo Zerlottin, Tommaso Bardelli, Mauro Dal Prà, Mattia Zuffada, Matteo Cremonesi, Luca Antonietti, Romana Bravi, Pier Giacomo Bianchi, Anna Pia Maria Giulini

**Affiliations:** 1Research Centre for Plant Protection and Certification, via Emilia km 307, 26838 Tavazzano con Villavesco, Italy; isabella.mazzoleni@crea.gov.it (I.M.);; 2Research Centre for Plant Protection and Certification, via G. Venezian 22, 20133 Milan, Italy; elena.novarina@crea.gov.it (E.N.); tommaso.bardelli@crea.gov.it (T.B.); piergiacomo.bianchi@crea.gov.it (P.G.B.); 3Università Cattolica del Sacro Cuore, Facoltà di Scienze Agrarie, Alimentari e Ambientali, via Emilia Parmense 84, 29122 Piacenza, Italy; yuki.zerlottin01@icatt.it; 4Research Centre for Plant Protection and Certification, via G. Marconi 2, 36045 Lonigo, Italy; mauro.dalpra@crea.gov.it (M.D.P.); romana.bravi@crea.gov.it (R.B.)

**Keywords:** maize, aflatoxins, biotic stress, climate change, new varieties

## Abstract

Plant growth and its interaction with microorganisms change yearly. High temperature and humidity have characterized recent seasons in the north of Italy and around the world, increasing the parasitic ability of *Aspergillus flavus* to colonize maize kernels and aflatoxin levels. These molecules have the highest acute and chronic toxicity of all mycotoxins; the maximal concentration in agricultural food and feed products, and their commodities, are regulated worldwide. In this study we suggest a simple methodology to test the susceptibility of candidate maize varieties to *A. flavus* before their release onto the market. A panel of 92 inbred lines and 14 hybrids were analysed, disease phenotypes were scored on artificially inoculated kernels using a rolled towel assay, and therefore we observed different responses to fungal infection on the kernels, outlining a high variability among the tested lines characterized by a different effect of the pathogen on seedling development. Even the hybrids responded differently on a statistical basis to *A. flavus* with regard to the development of coleoptile, allowing their categorization into classes of susceptibility to be used for the varietal registration. Interestingly, the hybrid 6a-A was less susceptible to *A. flavus* compared to its reciprocal in terms of the length of the coleoptile. The comparison of breeding lines released on the market in different years suggested a poor improvement in genetic resistance against *A. flavus* in maize so far, opening up a possible topic for future research aimed at mitigating the impact of climate change on agriculture.

## 1. Introduction

Maize (*Zea Mays* L.) is the most widely grown agricultural crop in the world that has become a model organism for basic and applied research in plant biology [1]. This dual importance of maize is largely due to its complex and diverse genome, which has allowed researchers to understand genetics, cytogenetics and genomics better, and has offered a rich pool of genetic diversity, helping breeders to improve germplasm [2].

This agricultural species, mainly used as a global feed, is also important as a food crop, particularly in sub-Saharan Africa and Latin America, in addition to being cultivated for other non-food uses such as industrial and energy crops [3]. The global maize area (for dry grain) stands at 204 M ha, and the annual production overtakes 1 billion metric tons [4]. The breeding programs, developed principally by private seed companies, are dynamic, releasing every year numerous maize hybrids characterized by better performance compared with all the material released on the market. The improved germplasm is crucial, not only for raising the potential yield, but also for addressing the emerging challenges which arise continuously due to climate change.

Maize is considered one of the crops most susceptible to mycotoxins world-wide [5]. Specifically, hot and humid climates may have an impact on maize contamination by several fungal species, mostly *Aspergillus, Alternaria, Fusarium* and *Penicillium* [6,7], and on the presence of mycotoxins. A recent analysis of European Food Safety Agency (EFSA) data reveals that up to 80% of the crops grown, stored and traded worldwide are contaminated with detectable quantities of secondary fungal metabolites classed as mycotoxins, with 20% being over the legal limit for contamination [8]. Among mycotoxins, aflatoxin B1 (AFB1) is one of the strongest human carcinogens [9]. *Aspergillus flavus* and *A. parasiticus* are the main producers of aflatoxins and, together with other species of *Aspergillus* section *Flavi*, have the capacity to generate this secondary metabolite [10]. The optimal growth of *A. flavus* occurs over the range of 19–35 °C [11], with 28 °C being optimum for aflatoxin production [12,13].

Maize is susceptible to infection and colonization by *A. flavus* and aflatoxin production during both the pre- and post-harvest phases of crop growth and storage. The contamination of aflatoxin increases from pre- to post-harvest [14], and it is strongly affected by temperature and moisture availability.

*A*. *flavus* can be divided into two distinct evolutionary lineages designated as 1B and 1C [15]. Lineage 1B strains, predominantly clonal, are mostly not aflatoxigenic, few of them can produce aflatoxins but in low amount. Conversely, the latter strains vary widely in their ability to produce aflatoxins [16]. *A. flavus* is ubiquitous and genetically and phenotypically very diverse [17]. This fungus can reproduce not only asexually by conidia and sclerotia [18], but also sexually through the interaction between mating type loci MAT1-1 and MAT1-2 [19], thus generating new genotypes [20].

The primary source of fungus inoculum is soil, specifically in the fields of highly susceptible crops, but also in forest ground without hosts, demonstrating its saprophytic ability [18]. Under favourable conditions, which include high temperature and water stress [21], the spores of the fungus, in the form of wind-dispersed conidia released from mycelium and sclerotia on soil surfaces, infect the developing inflorescences of maize. Once deposited on the silks [22], these spores colonize kernels’ surfaces and invade all the tissues of the seed through the pedicel region, through wounds created by insects or the mechanical injury of the pericarp [23,24,25], producing aflatoxin.

Histological studies performed on maize infected artificially by the conidia of *A. flavus* with a needle showed the localization of fungal mycelium and its morphological changes during the colonization of the diverse parts of the seed, taking into account that *A. flavus* preferentially colonizes tissues with high oil content as the germ. From endosperm, the fungus reached the germ through the basal transfer layer (BETL) of the basal endosperm and the scutellum [25] then, with intact aleurone layers, the fungus grows around the kernel between the pericarp and aleurone with a minimal colonization of the endosperm, reaching the scutellum and the germ from the aleurone layer which forms a barrier limiting the route of the fungus through the endosperm [26,27]. At the endosperm–germ interface, *A. flavus* often constitutes a biofilm-like structure surrounding the germ, and it is hypothesized that this structure contributes to the germ infection [23]. Fungal growth was slower or limited in the resistant hybrid compared to the susceptible ones, specifically in the aleurone layer or at the endosperm–scutellum interface [27]. Indeed, resistant hybrids may differentiate endosperm-specific antifungal proteins [28], hydrolytic enzymes [29,30] and antifungal ribosome-inactivation proteins in the scutellum [31].

Several strategies may be combined to produce healthy crops free of aflatoxin, even if increasing resistance seems to be the best solution [32].

The genetic basis of resistance to *Aspergillus flavus* has been studied using different maize populations, indicating that the resistance is a polygenic character in which additive effects play an essential role. The trait also showed low heritability [33,34], although higher heritability has been reported in the literature [35,36]. Large gene–environment interactions reduce the possibility of identifying good markers to select maize genotypes with a resistance to *A. flavus* [33,34,36,37,38,39,40,41,42,43,44,45,46,47]. Interestingly, Zhang and colleagues [46] combined linkage-based mapping on Quantitative Trait Locus (QTL) with Genome-Wide Association Studies (GWAS) to resolve a major QTL for *A. flavus* to identify several candidate genes for resistance. The meta-analysis of QTLs [48] gave good results in pinpointing candidate genes (and markers) associated with resistance to the fungus. Specifically, three markers close to MQTL2.4, MQTL4.1 and MQTL8.2 were identified for their great potential to be used in marker-assisted selection for *A. flavus* resistance. Furthermore, the identification of MQTLs allowed the authors to search for candidate genes with a potential involvement in maize–*A. flavus* interaction. For instance, a polyphenol oxidase (PPO) that plays an important role in plant defence mechanisms against biotic stresses, a plasma membrane intrinsic protein (PIP) that is involved in channels’ functioning, and an RNA-directed RNA polymerase that controls epigenetic changes and could act in disease resistance response have all been identified. Likewise, proteomic and transcriptomic analyses have helped to detect several proteins expressed during the infection of maize seeds by *A. flavus*. Few proteins are known to have antifungal activity, or to be involved in host defence. These include pathogenesis-related (PR) proteins, lipoxygenases, alpha-amylase inhibitors and ribosome-inactivating proteins [49].

Another possibility in ensuring maize hybrids to be more resistant to the *A. flavus* on the market is to revise the procedure for varietal registration, introducing new valuations to the final approval of the candidate varieties. Indeed, within the European Union (EU), crop varieties must be included in a Member State’s National List and/or common catalog before being marketed. This requires plant variety testing processes to evaluate whether the variety candidate is distinct, uniform and stable (DUS), and meets the requirement of value for cultivation or use (VCU). So far, the technical protocol for the approval of new maize varieties in Italy and in several European countries does not include the characterization of the resistance to *Aspergillus flavus* in candidate varieties.

In this context, the primary goal of this study was to phenotypically evaluate maize genotypes, part of our reference collection, for resistance to *A. flavus* using the RTA assay (Rolled Towel Assay), and to consider the effects of the artificial infection on seed germination to disclose a potential mechanism for resistance on mature kernels. The RTA has been previously and successfully applied to both soybean and maize genotypes to evaluate the resistance to *Fusarium* spp. [50,51,52,53]. So far, the genetic architecture of resistance to the *A. flavus* infection of seedlings has not yet been investigated using the Rolled Towel Assay. In this study, we described the different responses to artificial inoculation of mature maize kernels with spores of *A. flavus*, showing a variability in susceptibility to the fungus among the selected genotypes. The seedling development was determined between control vs. *A. flavus* inoculation in 77 maize lines and 14 hybrids using a five-point severity scale and measuring the length of the coleoptile.

As the final aim of this study, we proposed a procedure to categorize the hybrids in specific classes of susceptibility to the fungal infection to apply during the varietal registration process in order to evaluate the resistance of new hybrids. We also suggested measuring officially the resistance to *A. flavus*, not only in the hybrid but also in its reciprocal, once the preliminary data confirmed and highlighted the potential effect of the parental line on the tolerance to the fungus.

Appling the same methodology, we also tested the susceptibility to *A. flavus* in breeding lines released on the market in different years (2001–2010–2022), disclosing the breeding effort on resistance to *A. flavus* in maize along the years.

## 2. Results

### 2.1. Germination Scores and Disease Severity

The germination scores revealed that 41 out of 92 inbred lines had more than 90% of the germination in the control assay (Figure 1). The lines with less than 50% germination were discarded, leaving 77 lines for further analysis. No frequencies of maize lines were detected in either the 10% or 20% classes of germination (Figure 1).

The low germination observed in some inbred lines could be attributed mainly to two factors: the long-term storage of the kernels [54] and the sterilization procedure applied in the RTA protocol. Indeed, more than half of the selected lines were stored in a cold room for more than 10 years, and several lines displayed a reduction on the germination scores only if treated with ethanol and bleach that may affect the vitality of the kernel with non-visible damage on the surface.

Comparing the control and *A. flavus* inoculation of the 77 maize lines, we observed a reduction in percentage of the score 1 (healthy and germinated seedling with no visible sign of colonization; 67% vs. 8%), showing a relevant effect of the fungus infection that also appears with an increase in proportion of the score 2 (germinated seedling with slight colonization of the kernel near the pedicel; 11% vs. 59%) and 4 (germinated seedling with reduced development with complete colonization of the kernel; 0% vs. 4%). However, the percentage of score 5 (no germination; 19% vs. 26%) was quite similar between the control and inoculation (Figure 2), probably due to the damage of kernels or their contamination also recorded in the control assays, indicating a presence of infection occurring inside the kernel [55].

All the raw data concerning the severity of the infection (score), together with the phenotypic measurement determined in maize lines as the length of the coleoptile, were reported in Appendix A.

### 2.2. Effect of A. flavus Inoculation on the Length of Coleoptile in Selected Inbred Maize Lines of the Panel

The effect of inoculation with *Aspergillus flavus* on the length of coleoptile was performed on selected maize lines (public lines) of the panel, characterized by the higher percentage of germination and seed vigour. The length of the coleoptile, computed as the mean of ten kernels for each inbred line for the two replications carried out as a function of treatment (control vs. inoculation with *A. flavus*), is reported in Appendix A. Significant differences were found among lines and treatment, but not between replications, confirming the reproducibility of the RTA assay (Appendix A). According to the treatment (control vs. inoculation), specific and different responses were observed among the selected maize lines (Figure 3). The length of the coleoptile was negatively affected by the inoculation in five genotypes (Appendix A). In particular, the greatest reduction was observed in line B73 followed by Hi53, TSU-CHIAO-HSI-WU 102, INBRED 39-1546 and F252 (Figure 3). No significant difference between control and inoculation was recorded for the following lines: W22 BRINK, T143, CML182 and Mp705 (Figure 3). Among the lines studied, the length of the coleoptile in B73 was highly inhibited by *A. flavus* (Figure 4A), whereas a promoting growth in the inoculated kernels was found in CE-777 (Figure 4B). This latter result was previously observed by several authors [56,57], with *F. verticillioides* boosting the growth in germinating seeds as a defence mechanism.

Afterwards, significant differences among maize lines were further analysed only in the inoculated kernels to underline their susceptibility to *A. flavus* (Appendix A; Figure 5).

A more pronounced length of coleoptile was recorded in line Mp705, which seems to be less susceptible to *A. flavus* compared to the other genotypes (Figure 5). On the other hand, lines B73 and F252 showed the strongest reduction in seedling development, confirming the susceptibility of the B73 inbred line as described by several authors [58,59]. Furthermore, lines W22 BRINK, TSU-CHIAO-HSI-WU 102 and INBRED 39-1546 clustered together, resulting in no difference with lines T143, CML182 and CE-777. Line Hi53 was in between, with maize lines clustering in bc and d (Figure 5).

### 2.3. Effect of A. flavus Inoculation on the Length of Coleoptile in Maize Hybrids

The Rolled Towel Assay (RTA) was also performed on maize hybrids, with the final aim being to develop a methodology to apply within the registration process to evaluate the resistance to this fungus on the new candidates’ varieties. Seven hybrids and their reciprocal (a total of 14 hybrids) were counted, and the length of coleoptile computed as the mean of ten kernels per each hybrid, determined in two replications carried out as a function of treatment (control vs. inoculation), as shown in Appendix A. Significant differences were found among hybrids and treatment (Appendix A), but not between replications, confirming the reproducibility of RTA assay (Appendix A).

The seven hybrids, labelled with a specific code, responded statistically differently to *A. flavus* based on the development of coleoptile (Appendix A; Figure 6). Even if the number of studied hybrids is limited, it still allowed variations in terms of susceptibility to the fungal infection to be identified, pointing out three clusters. The hybrid labelled with code 6a showed the highest value, appearing to be more tolerant to the infection, followed by the hybrid 7a, and then both hybrids 4a and 5a as sensitive to *A. flavus*. The remaining three hybrids (1a, 2a and 3a) are not clearly distinguishable in specific groups, although they exhibit their own responses to the infection.

### 2.4. Effect of Parental Line on the Susceptibility to A. flavus in Maize Hybrid

To explore the impact of parental lines on the susceptibility of seven hybrids to *A. flavus*, we evaluated the effect of the inoculation in each hybrid and its reciprocal in terms of the length of the coleoptile. The analysis showed significant differences only in one hybrid 6a-A and its reciprocal 6a-B (Appendix A, Figure 7), highlighting a possible effect of parental lines on the susceptibility to the fungus on the hybrids.

### 2.5. Variations in the Inoculated Maize Lines According to the Year

Three years (2001 vs. 2010 vs. 2022) were considered to unravel possible variations in the inoculated maize lines with *A. flavus* (Appendix A). In detail, sixteen maize lines registered in the Italian National list were selected in each year, and the length of the coleoptile was determined as the mean of ten kernels per each line in two replications (Appendix A).

The length of the coleoptile significantly changed across years (Appendix A), recording higher values in the control compared to the inoculation maize lines (2010 and 2022) (Figure 8). In the year 2001, we did not observe any effect between the control and the inoculation (Appendix A). Lower values were recorded in the year 2001 compared to in 2010 and 2022, when less fluctuation among the means of detected maize lines (identified with points in Figure 9) was observed. Significant differences were found between 2001 and the remaining years (Figure 9), whereas the length of the coleoptile was similar between the years 2010 and 2022.

## 3. Discussion

The genetic basis of resistance to *Aspergillus flavus* is a polygenic character, in which the additive effects play an essential role. The maize resistance to the fungus is specific to different stages of pathogenesis (infection, colonization and reproduction) and effective in distinct tissues [60]. The trait also showed a large gene–environment interaction. Considering these specific features of the character, we studied the resistance to *A. flavus* in maize by inoculating mature kernels in vitro (fixed temperature and moisture) and applying the RTA assay. Accordingly, the results shown in this article refer to a specific tissue of the plant, revealing a potential mechanism for resistance in germinating seeds. The inbred lines analysed (10 kernels in two replications for 77 genotypes) showed multiple phenotypes underlying a different genotype–pathogen interaction. We classified the phenotypes using a five-point severity scale [52]. In some cases, the fungus was able to infect and colonize the kernel, easily suppressing the germination capacity of the seeds either partially (score 4) or totally (score 5). Additionally, a healthy and germinated seedling (score 1), a seedling with slight colonization of the kernel near pedicel (score 2) and another one with the widespread colonization of the kernel and browning of the coleoptile (score 3) were recorded. The spores of the fungus, loaded onto the kernel’s surface, were able to invade and colonize the seed differently among the tested genotypes. Previous studies have suggested several potential mechanisms in maize kernels to prevent fungus colonization and reproduction, such as grain texture [61], pericarp thickness or wax [62], intact aleurone, or dealing with the capacity to code endosperm-specific antifungal proteins [28], hydrolytic enzymes [29,30] and antifungal ribosome-inactivation proteins in the scutellum [31]. Our screening allowed us to select a set of inbred lines with high resistance to the fungus (score 1 and 2). This material could be characterized through further investigation to identify unexplored potential barriers in maize kernels evolved by plants against *A. flavus.*

Although similar kernels were selected with regard to size and shape, preferably flattened and without visible damage, we observed less vigour among kernels stored in the chamber since 2001. Indeed, even if seeds are stored in the most appropriate conditions, the vigour of seeds declines [63]. Therefore, we considered selected lines (public lines) of the panel for further analysis, characterized by a higher percentage of germination and seed vigour as they were produced in our field trial in 2023. All the materials, both lines and hybrids, were produced in the same field and year, which could be seen as a critical aspect considering that the environment in which the plant is grown could affect *A. flavus*–maize interaction [60].

We applied the RTA assay to study the effect of inoculation with *Aspergillus flavus* on a vegetative trait of maize, the length of the coleoptile, on 10 public lines of the panel. Significant differences were found among lines and treatment but not between replications, confirming the reproducibility of the RTA assay. Our results suggested that the fungus impacted on the length of the coleoptile differently among the genotypes tested, showing negative, positive and no effect on seedling development. The impaired growth in the inoculated kernels was previously observed by several authors [56,57], with *F. verticillioides* showing that the pathogen could affect the host fitness by promoting growth and inducing lignin deposition in the cell’s wall as a defence effort. The lines showing a positive or no effect on the seedling’s development could represent a valuable material to describe further and establish the mechanisms triggered for resistance to *A. flavus*. Interestingly, while observing line Mp705, it was found that this line seems to be less sensitive to *A. flavus* compared to the other genotypes in the RTA assay, and it was released in 1984 by USDA-ARS as a source of resistance to insect leaf-feeding damage but was considered susceptible to *Aspergillus flavus* under field conditions [47], reinforcing the concept of multiple components of resistance to *A. flavus* in maize. On the other hand, the observed line B73 showed the strongest reduction in a seedling’s development, confirming the susceptibility as it was described by several authors [58,59].

Once the robustness of the methodology on maize lines was confirmed, we applied the RTA assay to maize hybrids, which responded in a statistically different way to *A. flavus*, based on the development of coleoptile, showing a range of variability among our sample as the inbred lines. The variations spotted in our experiment, in terms of susceptibility to fungal infection, were pointed out in three clusters (labelled a, b and c, Figure 6). As expected for quantitative traits that change continuously, some of the hybrids analysed displayed a value in between the clusters (labelled ab and bc, Figure 6).

Considering the identified clusters, we would suggest four classes of susceptibility (T = tolerant, MT = moderately tolerant, MS = moderately sensitive, S = sensitive; Table 1) in order to categorize the hybrids during the varietal registration process in maize and to evaluate the resistance of hybrid candidates to *A. flavus* officially, both for organic and conventional varieties.

The RTA assay tested on maize kernels is a simple methodology, but at the same time it can disclose details about seed resistance to *A. flavus*, in addition to being adaptable to other fungi, like *Fusarium* spp. [53]. The varieties selected in this study will be explored to assess the resistance to the fungus in different tissues of the plant by applying different experimental conditions (laboratory vs. field). Considering the whole supply chain of maize, it would be of great importance to find a correlation between the data collected through the RTA assay and the capacity, in selected genotypes, to confine the fungus infection reducing aflatoxin production during storage.

As the varietal registration process could include the characterization of the hybrid and its reciprocal, we investigated the effect of the artificial inoculation of maize kernels in seven reciprocal crosses. Only the hybrid 6a-A resulted in less susceptibility to *A. flavus* compared to its reciprocal 6a-B (Figure 7) in terms of length of the coleoptile, assuming a parent effect on fungus resistance.

In plant breeding, along with combining the ability status of the parents, the maternal and reciprocal effects are crucial for the choice of inbred lines as female or male parents for several quantitative traits [64,65,66,67,68]. The maternal effect in plants has previously been studied and it could be related to the major contribution on cytoplasm for the nascent zygote and endosperm [69,70]. In maize, several parent-of-origin-effect loci screenings for seed mutants have been identified and their study has revealed altered grain fill [71,72,73] or aberrant endosperm development [74].

Relating to our preliminary result on hybrid 6a-A, more investigation is required to establish the extent to which parent effects influence the resistance to *A. flavus* in a maize kernel before implementing the test in the varietal registration process. Accordingly, we have planned to constitute a larger population of maize hybrids crossing the female line of the hybrid 6a-A with some selected inbred lines of our reference collection with the final attempt to phenotype the generated hybrids along with the parents with the RTA assay.

In conclusion, we compared the fungus effect on breeding lines released on the market in different years: as the length of the coleoptile significantly changed between the years 2001, 2010 and 2022, both in the control and treatment samples, we can speculate that the variation observed in 2001 could be due to both the quality of kernels that affect the germination capacity to develop the coleoptile and the susceptibility to the fungus. However, between the 2010 and 2022 lines, we found no significant differences in relation to the coleoptile length in treated samples, suggesting a poor contribution of breeding to reduce fungus invasion and toxin accumulation in maize kernels, making it vital to explore new ways to release resistant hybrids onto the market.

## 4. Materials and Methods

### 4.1. Maize Genotypes

A set of 106 maize genotypes (92 inbred lines and 14 hybrids) were selected.

Among the inbred lines, 10 were public lines: Mp705, W22 BRINK, Hi53, T143, TSU-CHIAO-HIS-WU 102, BRED 39-1546, CE-777, CML 182 were part of the collection of the U.S. Department of Agriculture (USDA); F252 and B73 were part of the reference collection of CREA DC. The outcrossing of the public lines was carried out in the summer of 2023 at CREA DC station (Tavazzano—LO). The seeds produced were used in the RTA assay. The hybrids (A × B) and their reverse (B × A), released by a private company, were labelled with a specific code shown in the Results section.

The germination test was conducted in a growth chamber at 28 °C and 95% humidity for five days in the dark. Of the 106 genotypes, 77 inbred lines and 14 hybrids, with more than 50% germination, were considered for further analysis.

### 4.2. Fungal Strain

Ten infected maize samples were collected during the year 2021, mainly from maize fields in Northern Italy (the Veneto region) by the CREA DC of Lonigo. Every sample consisted of two ears presenting macroscopic signs of *Aspergillus* colonization. Four hundred seeds per sample were surface-disinfected for 30 s in a 1% sodium hypochlorite solution, rinsed in sterile water, then placed in 60-mm Petri dishes containing Potato Dextrose Agar (PDA) medium amended with streptomycin sulphate (200 mg/L). All recovered *Aspergillus* colonies were single spore purified and then identified based on their morphological features, both macroscopically (mycelium texture and colour) and by stereomicroscope observation (presence of conidial heads). To confirm the morphological identification, every fungal colony was subjected to a genomic DNA extraction with the Quick-DNA Fungal/Bacterial Miniprep kit (Zymo Research, Beijing, China). The region of Internal Transcribed Spacer (ITS) was amplified using the primer pair ITS1/ITS4 [75]. The molecular identification of the strains was achieved by querying ITS sequences against the GenBank database. Three strains were identified as *A. flavus*. One was named MCL66, stored in the Lonigo specimen collection and selected as a source of inoculum. Conidia suspension was prepared by scraping the surface of two-week-old colonies grown on PDA with an L-shaped spreader and using double distilled sterile water. The suspension was filtered, and conidia concentration was determined by hemocytometer cell counting. The suspension was diluted at a concentration of 1 × 10^6^ mL^−1^ spore using double distilled sterile water and stored at 4 °C.

### 4.3. Experimental Procedure

Based on the protocol previously described [51,52,53,76], some modifications were applied. Seeds with a similar size and shape, and without visible damage, were selected, as indicated by [76]. The seeds for each genotype were surface-sterilized in a solution of 70% ethanol and shaken for 5′ at 100 rpm to reduce seed-borne contaminations. Then, seeds were washed by sterilized bi-distilled water for 1′, and by a 25 mL commercial bleach solution for 10′ at 100 rpm. Finally, seeds were rinsed three times (5′ each at 100 rpm) with sterilized bi-distilled water.

For each genotype, forty seeds were selected: 20 to be used as a treated sample and 20 to be used as a control.

Two Rolled Towel Assay (RTAs), for control and treated conditions, were prepared for each genotype. Three towels of germination paper (Anchor Paper, Saint Paul, MN, USA) for each RTA were moistened with sterilized bi-distilled water; ten seeds for the two replicates were placed and evenly spaced at 5–10 cm from the top edge on two base towels, and they were covered with the third towel marked with the maize genotype name, code C or A (Control or Treated) with *Aspergillus*, respectively, and the number of replication.

Control RTAs were inoculated with a 50 µL of sterilized bi-distilled water; in the treated RTA, seeds were inoculated with a 50 µL of a 1 × 10^6^ mL^−1^ spore suspension of *A. flavus*.

After the inoculation, the towels were rolled up, placed vertically in a bucket and kept in groups in transparent plastic bags separately for treating and control to avoid cross-contamination [51]. RTAs were incubated in a growth chamber at 28 °C with 95% humidity, in the dark for seven days. This temperature chosen in our experiment resulted in being optimal for both fungal growth and the germination of maize seeds to obtain a robust coleoptile.

### 4.4. Disease Severity Evaluation After Inoculation

After incubation, the severity of the inoculation and the control assays were evaluated using a five-point severity scale [51,52,53] that we optimized according to our purpose; in detail: 1 = healthy, germinated seedlings with no visible signs of colonization; 2 = germination with slight colonization of the kernel near the pedicel; 3 = germination with widespread colonization of the kernel and browning of the coleoptile; 4 = germination with reduced seedling development, with complete colonization of the kernel; 5 = no germination. Afterwards, the length of the coleoptile was measured with a ruler for each kernel.

### 4.5. Statistical Analysis

Statistical analyses were performed with the Software XLSTAT (version 4.1.2022). Prior to performing the ANOVA analysis, the normality of the dataset was verified using the Shapiro–Wilk or Kolmogorov–Smirnov tests; a log transformation of the length of the coleoptile in B73 line was required.

The analysis of variance (ANOVA) was performed to evaluate the effect of *A. flavus* in inoculated vs. control kernels, and to determine different responses among the maize genotypes in terms of fungal resistance. Significant differences (*p* > 0.05) were recorded and analysed by paired comparison with the Fisher post-hoc test.

## 5. Conclusions

In this study, a total of 92 inbred lines and 14 hybrids of maize were analysed to unravel different responses to artificial inoculation with spores of *A. flavus*, through the germination test (RTA). The results suggested that the fungus affected the tested genotypes differently, showing negative, positive and no effect on seedling development. The methodology applied to the hybrids allowed us to point out three clusters of susceptibility to fungal infection. The further step, considering the established clusters, was to identify four classes of susceptibility (T = tolerant, MT = moderately tolerant, MS = moderately sensitive, S = sensitive) in order to categorize the hybrids during the varietal registration process in maize and officially to select hybrids that are more resistant to *A. flavus*. Indeed, the improved germplasm is crucial, not only to raise the potential yield, but also to address the emerging challenges which arise continuously due to climate change.

Furthermore, this assay allowed us to discover a parent effect on the resistance to *A. flavus* in maize kernels, and we observed that one hybrid showed less susceptibility to the fungus over its reciprocal in terms of the length of the coleoptile.

Overall, we encourage the methodology used in this study to be applied to candidate maize hybrids during the varietal registration process as a tool to release material that is more resistant to the fungus onto the market.

## Figures and Tables

**Figure 1 plants-14-01109-f001:**
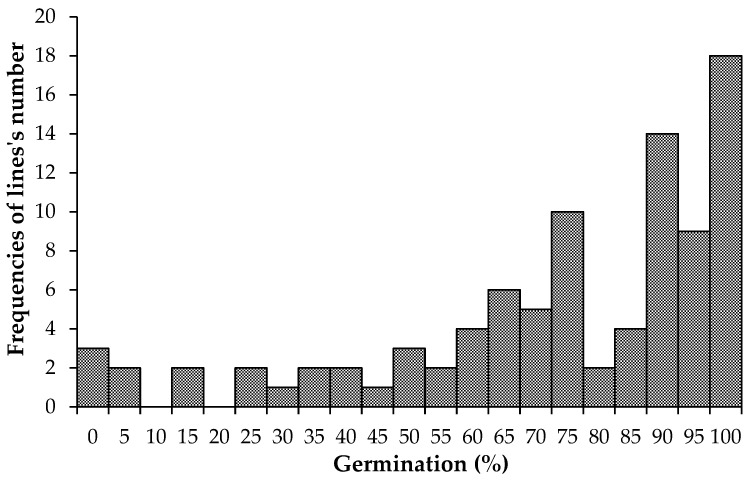
The percentage of germination on maize lines tested in the control assay.

**Figure 2 plants-14-01109-f002:**
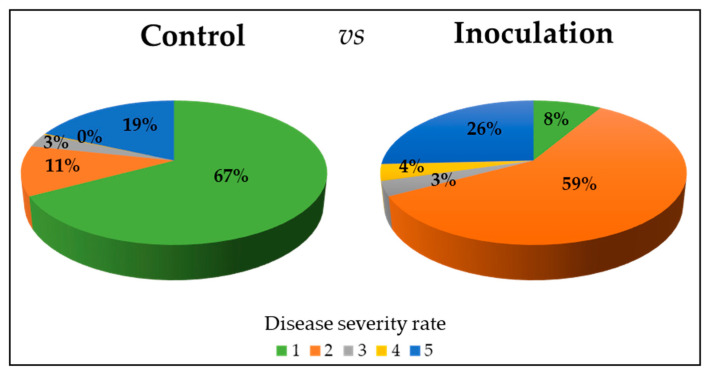
Disease severity rate in the control vs. inoculation rolled towel assays of kernels in 77 maize lines. The seedling development was determined using a five-point severity scale as previously described in [52] (1 = healthy and germinated seedling; 2 = germinated seedling with slight colonization of the kernel near pedicel; 3 = germinated seedling with widespread colonization of the kernel and browning of the coleoptile; 4 = germinated seedling with reduced development and complete colonization of the kernel; 5 = no germination due to complete rotting of the kernel).

**Figure 3 plants-14-01109-f003:**
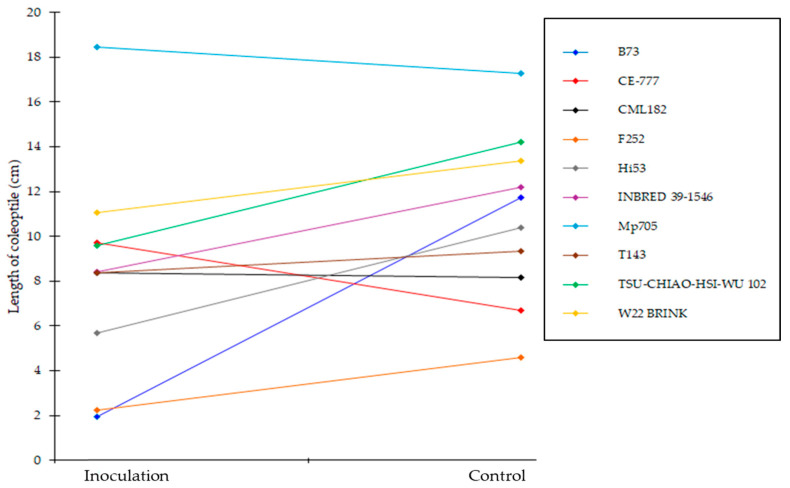
Effect of the treatment (inoculation with *A. flavus* vs. control) on the length of the coleoptile along different maize lines.

**Figure 4 plants-14-01109-f004:**
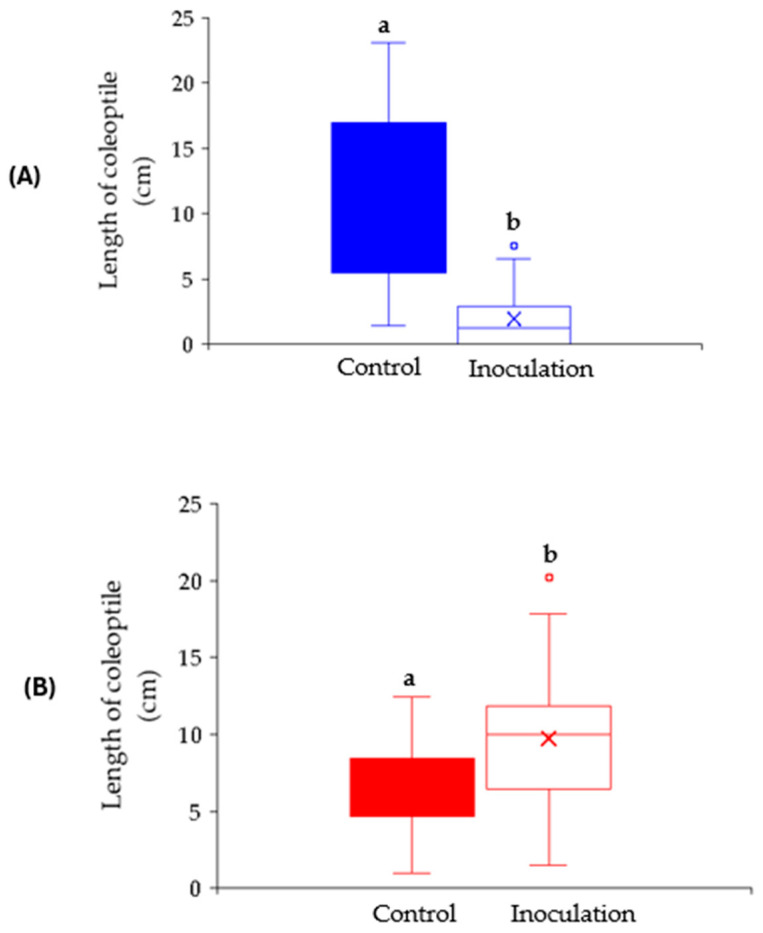
Length of the coleoptile observed in the control vs. inoculated kernels in line B73 (**A**) and in line CE-777 (**B**). Different letters indicate significant differences (*p* ≤ 0.05; ANOVA followed by the Fisher post-hoc test) as a function of the treatment.

**Figure 5 plants-14-01109-f005:**
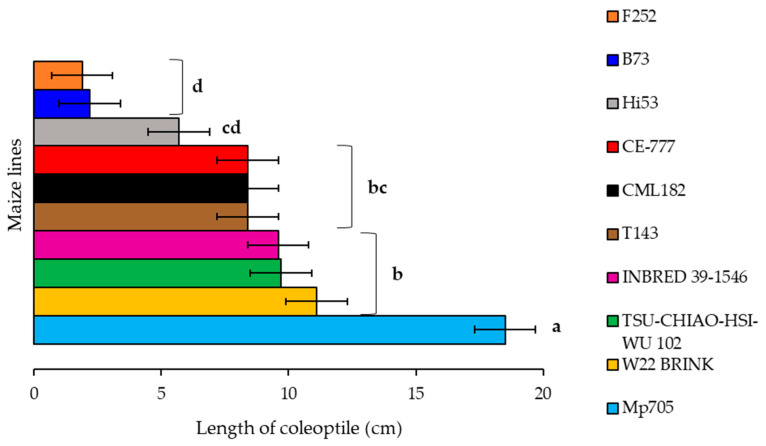
Comparisons between selected maize lines with regard to *A. flavus*, considering the length of the coleoptile. Different letters indicate significant differences (*p* ≤ 0.05; ANOVA followed by the Fisher post-hoc test) as a function of maize lines.

**Figure 6 plants-14-01109-f006:**
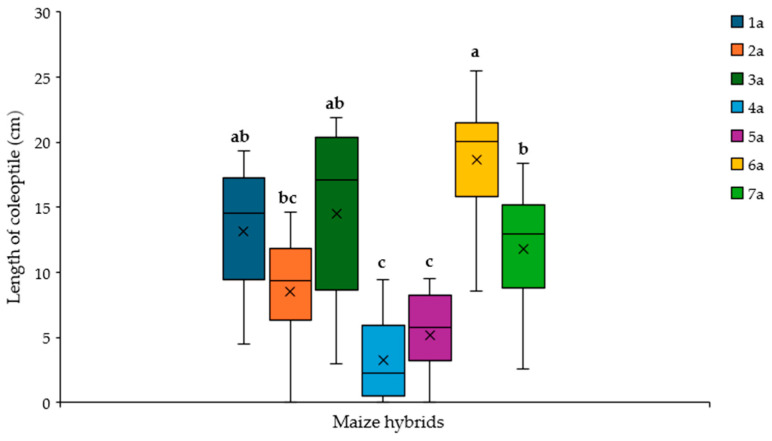
Length of the coleoptile measured in the inoculated kernels of selected maize hybrids labelled with a specific code (from 1 to 7). Different letters indicate significant differences (*p* ≤ 0.05; ANOVA followed by the Fisher post-hoc test) as a function of the hybrids.

**Figure 7 plants-14-01109-f007:**
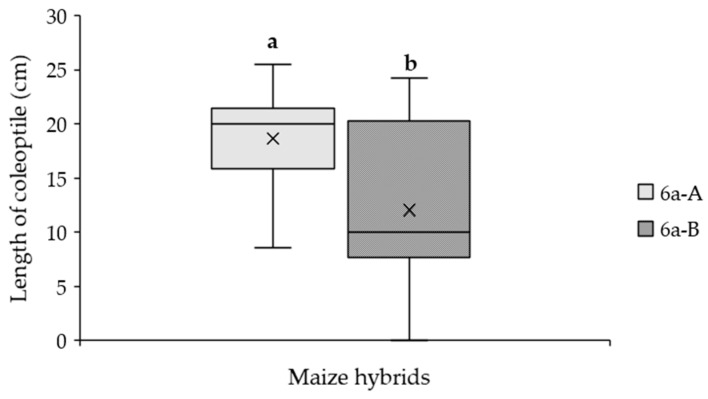
Length of the coleoptile measured in the inoculated kernels in a specific maize hybrid (6a-A) and its reciprocal (6a-B). Different letters indicate significant differences (*p* ≤ 0.05; ANOVA followed by the Fisher post-hoc test) as a function of the hybrid.

**Figure 8 plants-14-01109-f008:**
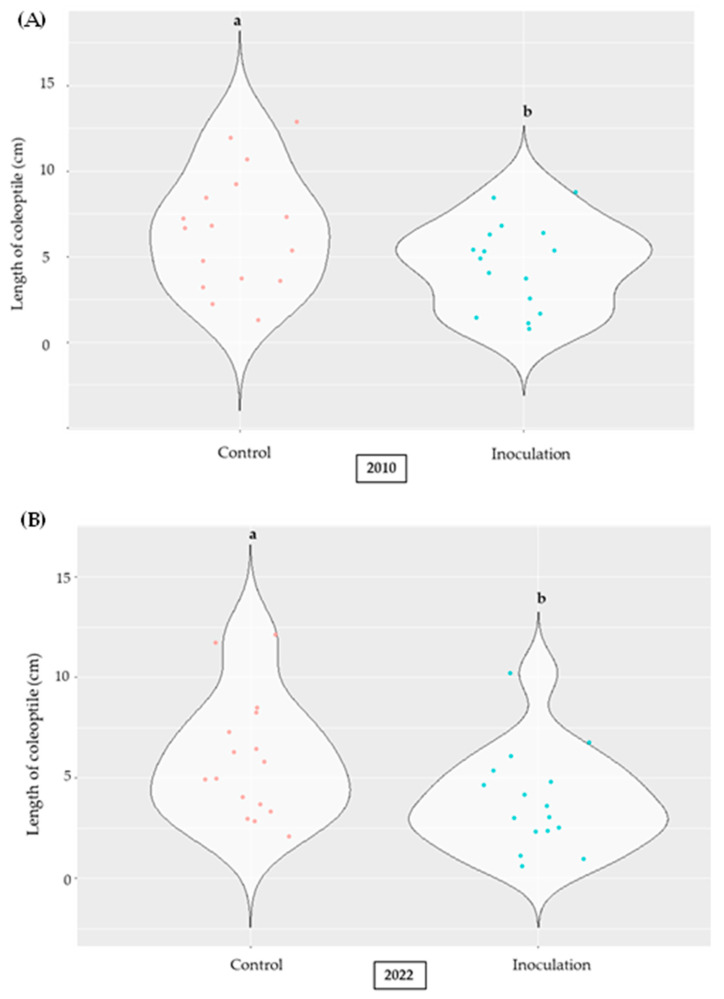
The violin plots show the length of the coleoptile determined in the control vs. inoculated kernels in 32 selected maize lines (16 genotypes × 2 replications) in 2010 (**A**) and 2022 (**B**). The length of the coleoptile was computed as the mean of ten kernels in each replication; different letters indicate significant differences (*p* ≤ 0.05; ANOVA followed by the Fisher post-hoc test) as a function of treatment.

**Figure 9 plants-14-01109-f009:**
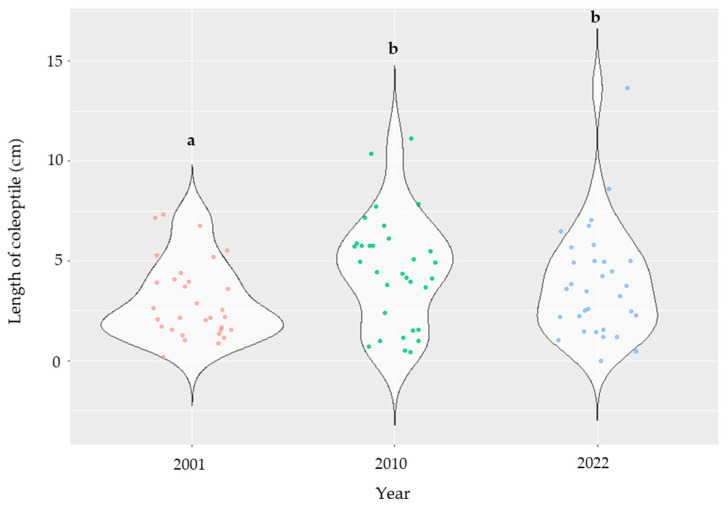
The violin plots show the length of the coleoptile determined in the inoculated kernels in 32 selected maize lines (16 genotypes × 2 replications) across years (2001 vs. 2010 vs. 2022). The length of the coleoptile was computed as the mean of ten kernels in each replication; different letters indicate significant differences (*p* ≤ 0.05; ANOVA followed by the Fisher post-hoc test) as a function of year.

**Table 1 plants-14-01109-t001:** Classes of susceptibility to the fungal infection.

CLASS	T	MT	MS	S
CLUSTER	a	b	c	d

T = tolerant, MT = moderately tolerant, MS = moderately sensitive, S = sensitive.

## Data Availability

Data are contained within the article.

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
