# Peer review of "The Effect of *Aspergillus flavus* on Seedling Development in Maize"

_plants, 2025, doi:10.3390/plants14071109_

Round 1

Reviewer 1 Report

Comments and Suggestions for Authors

The reference [75] at line 425 seems to be meaningless, therefore a throughout revision of all the links between statements and references is absolutely needed.

References in the 4.3 paragraph are not numbered. After numbering, the reference list is likely to be revolutionized, therefore, again, a throughout revision of all the links between statements and references is absolutely needed.

Since the optimal growing conditions for A. flavus are 19-35°C and 28°C is the optimum temperature just for aflatoxin production, authors should better explain why they choose 28°C for the germination test.

Comments on the Quality of English Language

The authors use sometimes the word “Interesting” as an incipit, while the correct word is possibly “Interestingly”.

In the paragraph 4.2, “Aspergillus flavus“ and “A. flavus “ should be written in italics.

Author Response

Reviewer 1

Revision Report 

General comments

The article deals with the assay of the effect of Aspergillus flavus on seedlings of different maize genotypes.

This argument is worth to be investigated since A. flavus infections heavily impact on kernel quality and food safety.

There are heavy issues in the paper that make the paper itself not acceptable for publication.

We thank the reviewer for his/her comment, and we improved our paper accordingly.

Language

As a whole, the paper is well-written.

We thank the reviewer for his/her positive comment.

Language

As a whole, the paper is well-written.

The authors use sometimes the word “Interesting” as an incipit, while the correct word is possibly

“Interestingly”.

In the paragraph 4.2, “Aspergillus flavus“ and “A. flavus “ should be written in italics.

Response: We have, accordingly, revised the paper in terms of English editing. In detail, we changed “Interesting” as correct word “Interestingly” through the manuscript. Moreover, we wrote “A. flavus” in italics in paragraph 4.2.

Experimental design

The experimental design followed in the paper is rather consolidated and the test approach that can be considered as a standard.

We thank the reviewer for his/her positive comment.

Under the methodological point of view, the authors report that the optimal growing conditions for A. flavus are 19-35°C whereas 28°C is the optimum temperature just for aflatoxin production, authors should therefore better explain why they choose 28°C for the germination test.

Do they mean that optimum temperature for aflatoxin production is the best also for infection?

This point is crucial and need to be better explained.

Done, we added more details about the methodology on the growing conditions of A.flavus. Please find them in the text at line 453-455.

Results and discussion

The conclusions are consistent with the evidence and they address the basic question posed by the paper.

We thank the reviewer for his/her positive feedback.

References

The reference [75] at line 425 seems to be meaningless, therefore a throughout revision of all the links between statements and references is absolutely needed.

We agree with this comment, and we revised the link of the reference [75] and the others through the manuscript.

References in the 4.3 paragraph are not numbered. After numbering those, the whole reference list is likely to be revolutionized, therefore, again, a throughout revision of all the links between statements and references is absolutely needed.

We numbered the references in 4.3 paragraph, and we revised all the links of the reference through the manuscript.

We added the following new references:

  1. Yao, W.H.; Zhang, Y.D.; Kang, M.S.; Chen, H.; Li, L.; Yu, L.J.; Fan, X. Diallel analysis models: a comparison of certain genetic statistics. Crop Sci 2013, 53, 1481-1490. https://doi.org/10.2135/cropsci2013.01.0027
  2. Fan, X.M.; Zhang, Y.D.; Yao, W.H.; Bi, Y.Q.; Liu, L.; Chen, H.M.; Kang, M.S. Reciprocal diallel crosses impact combining ability, variance estimation, and heterotic group classification. Crop Sci. 2014, 54, 89-97. https://doi.org/10.2135/cropsci2013.06.0393
  3. Fan, X.M.; Bi, Y.; Zhang, Y.; Jeffers, D.; Yin, X.F.; Kang, M. Improving breeding efficiency of a hybrid maize program using a three heterotic-group classification. J. Agron. 2018, 110, 1209-1216. https://doi.org/10.2134/agronj2017.05.0290
  4. Dosho, B.M.; Ifie, B.E.; Asante, I.K.; Danquah, E.Y.; Zelekhe, H. Combining ability of quality protein maize inbred lines under low and optimum soil nitrogen environments in Ethiopia. Afr. J. plant Sci. 2021, 15, 237-249. https://doi.org/10.5897/AJPS2021.2145
  5. Bonipas, A. J.; Rajashekhar, M. K.; Gopalakrishna, N.; Sidramappa, C. T.; Zerka, R.; Bindiganavile, S.V.; Nagesh, P.; Shiddappa, R.S.; Prema G.U. Maternal effects, reciprocal differences and combining ability study for yield and its component traits in maize (Zea mays L.) through modified diallel analysis. Peer J. 2024, 12, 17600. https://doi.org/10.7717/peerj.17600

Reviewer 2 Report

Comments and Suggestions for Authors
  • You ought to be more careful with the language since the way this article is currently written conspires against a fluent reading and understanding of the message you are trying to convey.
  • To improve the future presentation of a rewritten version of this article, please note that in Fig. 1, you mention 'germination rates' when referring to 'germination %', as the horizontal label indicates. The class 10% is missing from the chart.
  • The following assertion, “Future research should include histological and chemical characterization of the grains of the genotypes studied with extreme phenotypes to identify unexplored potential barriers evolved by the plant against A. flavus”, points to a potential lack of reference searching and/or knowledge of the plant/fungus pathosystem under analysis.
  • On the other hand, grandiloquent sentences like “The parent-of-origin effects may be induced by loci required for gametophyte development or by imprinted genes involved in seed development. The maternal effect in plants was previously studied and it could be related to the major contribution on cytoplasm for the nascent zygote and endosperm [66, 67]. In addition, the central cell genome has more open chromatin giving a dominant maternal role to initiate the coordinate development of the endosperm and embryo” neither has nothing to do with the work performed nor adds to the interpretation of the very simple results shown in this draft.
  • All in all, this research article must be improved to demonstrate why it should be considered relevant to a community of professionals already aware of such an important crop with a huge history of horticultural intervention, phytopathological studies and improvement methods.
Comments on the Quality of English Language

The article is easy to follow, but the Italian syntax sometimes makes it difficult to understand what the authors meant. Some words were not checked before submitting the article, e.g., ‘contest’ instead of ‘context’, ‘germinations score’ instead of ‘germination scores’, “displaied” instead of “displayed”, and many others. If sent again, the text needs to be revised by a native English speaker.

Author Response

Reviewer 2

Comments and Suggestions for Authors

You ought to be more careful with the language since the way this article is currently written conspires against a fluent reading and understanding of the message you are trying to convey.

We improved the English in order to make more fluent the reading and the message we want to convey. The text was revised by our colleague having expertise in the English editing considering that the technical procedure of our Research Centre does not allow to lean on the professional support (MDPI Author Services) to the language editing offered by the Journal.

To improve the future presentation of a rewritten version of this article, please note that in Fig. 1, you mention 'germination rates' when referring to 'germination %', as the horizontal label indicates. The class 10% is missing from the chart.

Done. We revised Figure 1 according to the reviewers’ comments.

The following assertion, “Future research should include histological and chemical characterization of the grains of the genotypes studied with extreme phenotypes to identify unexplored potential barriers evolved by the plant against A. flavus”, points to a potential lack of reference searching and/or knowledge of the plant/fungus pathosystem under analysis.

We agree with this comment, and we revised the assertion. Please find it in the text at lines 313-316.

On the other hand, grandiloquent sentences like “The parent-of-origin effects may be induced by loci required for gametophyte development or by imprinted genes involved in seed development. The maternal effect in plants was previously studied and it could be related to the major contribution on cytoplasm for the nascent zygote and endosperm [66, 67]. In addition, the central cell genome has more open chromatin giving a dominant maternal role to initiate the coordinate development of the endosperm and embryo” neither has nothing to do with the work performed nor adds to the interpretation of the very simple results shown in this draft.

We considered this comment, and we revised the assertion. Please find it in the text at lines 369-387.

All in all, this research article must be improved to demonstrate why it should be considered relevant to a community of professionals already aware of such an important crop with a huge history of horticultural intervention, phytopathological studies and improvement methods.

We agree with this comment, and we added more assertions to demonstrate the interesting point of view on the topic. Please find them in the text at lines 125-133; 141-145; 146-151; 313-316; 476-485.

Comments on the Quality of English Language

The article is easy to follow, but the Italian syntax sometimes makes it difficult to understand what the authors meant. Some words were not checked before submitting the article, e.g., ‘contest’ instead of ‘context’, ‘germinations score’ instead of ‘germination scores’, “displaied” instead of “displayed”, and many others. If sent again, the text needs to be revised by a native English speaker.

We thank the reviewer for his/her comment, allowing us to improve the English editing of our paper. Some English words were corrected accordingly.

Round 2

Reviewer 2 Report

Comments and Suggestions for Authors

I appreciate the effort made by the authors to improve the presentation of their results in this new draft. However, I think that some editing is still necessary, but I assume that this will be done during the editing process, if the article is finally approved for publication by the Journal Editor.

Comments on the Quality of English Language

The text improved very much, but some editing is still required.

Author Response

Dear reviewer,

thank you for the positive comment and we improved the new version adding a better explanation of the results in the conclusions.

Best regards